# Distilling Object Detectors with Feature Richness

Zhixing Du [1,2,3]     Rui Zhang [2,3] *     Ming Chang [3]
Xishan Zhang [2,3]     Shaoli Liu [3]     Tianshi Chen [3]     Yunji Chen [2,4]

[1]University of Science and Technology of China
[2]SKL of Computer Architecture, Institute of Computing Technology, CAS, Beijing, China
[3]Cambricon Technologies, China
[4]University of Chinese Academy of Sciences, China

dzx1@mail.ustc.edu.cn
{zhangrui,zhangxishan,cyj}@ict.ac.cn
{changming,liushaoli,tchen}@cambricon.com

## Abstract

In recent years, large-scale deep models have achieved great success, but the huge computational complexity and massive storage requirements make it a great challenge to deploy them in resource-limited devices. As a model compression and acceleration method, knowledge distillation effectively improves the performance of small models by transferring the dark knowledge from the teacher detector. However, most of the existing distillation-based detection methods mainly imitating features near bounding boxes, which suffer from two limitations. First, they ignore the beneficial features outside the bounding boxes. Second, these methods imitate some features which are mistakenly regarded as the background by the teacher detector. To address the above issues, we propose a novel Feature-Richness Score (FRS) method to choose important features that improve generalized detectability during distilling. The proposed method effectively retrieves the important features outside the bounding boxes and removes the detrimental features within the bounding boxes. Extensive experiments show that our methods achieve excellent performance on both anchor-based and anchor-free detectors. For example, RetinaNet with ResNet-50 achieves 39.7% in mAP on the COCO2017 dataset, which even surpasses the ResNet-101 based teacher detector 38.9% by 0.8%. Our implementation is available at https://github.com/duzhixing/FRS.

## 1   Introduction

Owe to the widespread use of deep learning, object detection method have developed relatively rapidly. Large-scale deep models have achieved overwhelming success, but the huge computational complexity and storage requirements limit their deployment in real-time applications, such as video surveillance, autonomous vehicles. Therefore, how to find a better balance between accuracy and efficiency has become a key issue. Knowledge Distillation [12] is a promising solution for the above problem. It is a model compression and acceleration method that can effectively improve the performance of small models under the guidance of the teacher model.

For object detection, distilling detectors through imitating all features is inefficient, because the object-irrelevant area would unavoidably introduce a lot of noises. Therefore, how to choose the important features beneficial to distillation remains an unsolved issue. Most of the previous distillation-based detection methods mainly imitate features that overlap with bounding boxes (*i.e.* ground truth objects), because they believe the features of foreground which can be selected from bounding boxes are

---

*Corresponding author

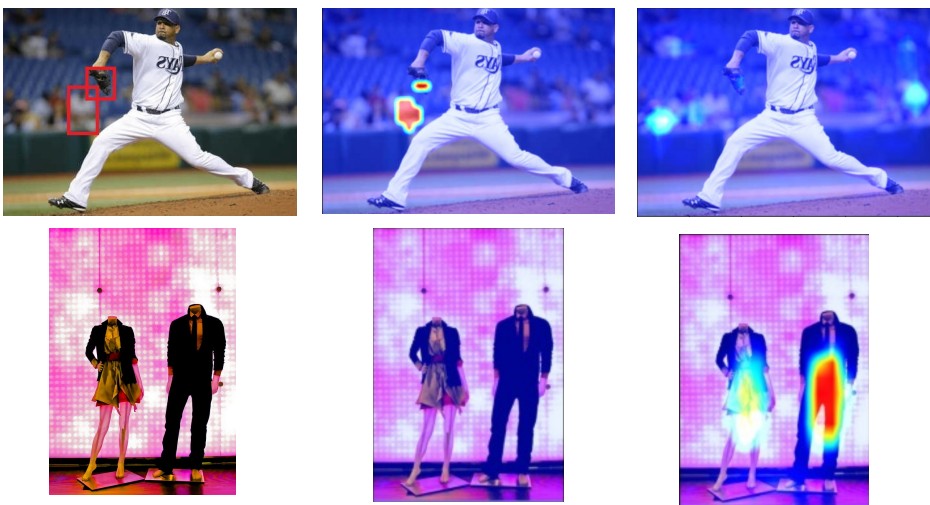

|(a) Images and bounding boxes|(b) Bounding box based method|(c) Ours|

Figure 1: Visualization of feature masks used in distillation of bounding box based method and ours.

important. However, these methods suffer from two limitations. First, foreground features selected from bounding boxes only contain categories in the dataset, but ignore the categories of objects outside the dataset, which leads to the omission of some important features, as shown in Figure 1 (b)-bottom. For example, the mannequin category is not included in the COCO dataset, but the person category is included. Since mannequins are visually similar to persons, the features of mannequins contain many useful characteristics of persons which are beneficial for improving the detectability of person for detectors in distillation. Second, only using the prior knowledge of bounding boxes to select features for distillation ignores the defects of the teacher detector. Imitating features that are mistakenly regarded as the background by the teacher detector will mislead and damage the distillation, as shown in Figure 1 (b)-top.

To handle the above problem, we propose a novel Feature-Richness Score (FRS) method to choose important features that are beneficial to distillation. Feature richness refers to the amount of object information contained in the features, meanwhile it can be represented by the probability that these features are objects. Distilling features that have high feature richness instead of features in bounding boxes can effectively solve the above two limitations. First, features of objects whose categories are not included in the dataset have high feature richness. Thus, using feature richness can retrieve the important features outside the bounding boxes, which can guide the student detector to learn the generalized detectability of the teacher detector. For example, features of mannequins that have high feature richness can promote student detector to improve its generalized detectability of persons, as shown in Figure 1 (c)-bottom. Second, features in the bounding boxes but are misclassified with teacher detector have low feature richness. Thus, using feature richness can remove the misleading features of the teacher detector in the bounding boxes, as shown in Figure 1 (c)-top. Consequently, the importance of features is strongly correlated with the feature richness, namely feature richness is appropriate to choose important features for distillation. Since the classification score aggregating of all categories is an approximation of probability that the features are objects, we use the aggregated classification score as the criterion for feature richness. In practice, we utilize the aggregated classification score corresponding to each FPN level in teacher detector as the feature mask which is used as feature richness map to guide student detector, in both the FPN feature and the subsequent classification head.

Compared with the previous methods which use prior knowledge of bounding box information, our method uses aggregated classification score of the feature map as the mask of feature richness, which is related to the objects and teacher detector. Our method offers the following advantages. First of all, the mask in our method is pixel-wise and fine-grained, so we can distill the student detector in a more refined approach to promote effective features and suppress the influence of ineffective features of teacher detector. Besides, our method is more suitable for detector with the FPN module, because our method can generate corresponding feature richness masks for each FPN layer of student detector based on the features extracted from each FPN layer of teacher detector. Finally, our method

is a plug-and-play block to any architecture. We implement our approach in multiple popular object detection frameworks, including one-stage, two-stage methods and anchor-free methods.

To demonstrate the advantages of the proposed FRS, we evaluate it on the COCO dataset on various framework: Faster-RCNN [24], Retinanet [18], GFL [17] and FCOS [27]. With FRS, we have outperformed state-of-the-art methods on all distillation-based detection frameworks. This achievement shows that the proposed FRS effectively chooses the important features extracted from the teacher.

## 2 Related Works

### 2.1 Object Detection

In recent years, noticeable improvements in accuracy have been made in object detection. The current mainstream object detection algorithms are mainly divided into two-stage detectors [24, 2, 9] and one-stage detectors [22, 17, 27, 18, 23]. Two-stage methods such as FasterRCNN [24] region proposal network and refinement procedure of classification and location. Although detectors fitted with very deep backbone have better detection accuracy, they are expensive in terms of computation cost and hard to deploy to real-time applications. While one-stage methods, such as Retinanet [18] and FCOS [27], maintain an advantage in speed that can directly generate the category probability and position coordinate value of the object. In object detection, the most anchor-based models [18, 24] have achieved good results by using predefined anchors, but the predefined anchors will bring a large amount of output and adjustments to the positions and sizes corresponding to different anchors. The anchor-free methods [27, 14, 6] is lighter, getting rid of redundant hyper-parameters by detecting an object bounding box as a pair of key points such as center and distance to boundaries. Although large-scale deep neural networks have achieved great success, the huge computational complexity and storage requirements make it a great challenge to deploy in real-time applications, especially on video surveillance and other resource-limited devices. The model compression and acceleration techniques, such as quantized [34] and knowledge distillation [12], become an important way to solve this problem.

### 2.2 Knowledge Distillation

Knowledge distillation is an effective method of model compression, which can improve the performance of small models. Hinton et al. [12] proposed to transfer the dark knowledge from teacher network to student network through the soft outputs which opens the door to the rapid development of knowledge distillation. There are three categories of knowledge for knowledge distillation [7]. Response-Based methods use the logits of a large deep model as teacher knowledge [12, 13, 21, 1, 33]. Feature-Based methods think that the features of intermediate layers also play an important role in guiding the learning of the student detector [25, 31, 11]. And Relation-Based methods believe that the relationship between different activations, neurons or sample pairs contains rich information learned by the teacher model [29, 30, 15, 20].

Recently, there are several methods which are used to compress object detectors by knowledge distillation. Chen et al. [3] proposed to distills the student detector through all components that ignore the imbalance in foreground and background, leading to poor results. Wang et al. [28] pay more attention to the near ground truth area. Li et al. [16] address the problem by distilling the positive and negative instances in a certain proportion sampled by RPN. Ruoyu et al. [26] propose to use a two-dimensional Gaussian mask to suppress irrelevant redundant background while emphasizing the target information. Dai et al. [5] propose a distillation method for detection tasks based on discriminative instances without considering the positive or negative distinguished by ground truth. Guo et al. [8] point out that the information of features derived from regions excluding objects is also essential for distilling the student detector, which did not explore the background area in more detail. However, we find the feature-rich areas outside the bounding box regions also play an important role in knowledge distillation. It is unreasonable to extract features by simply distinguishing positive and negative samples.

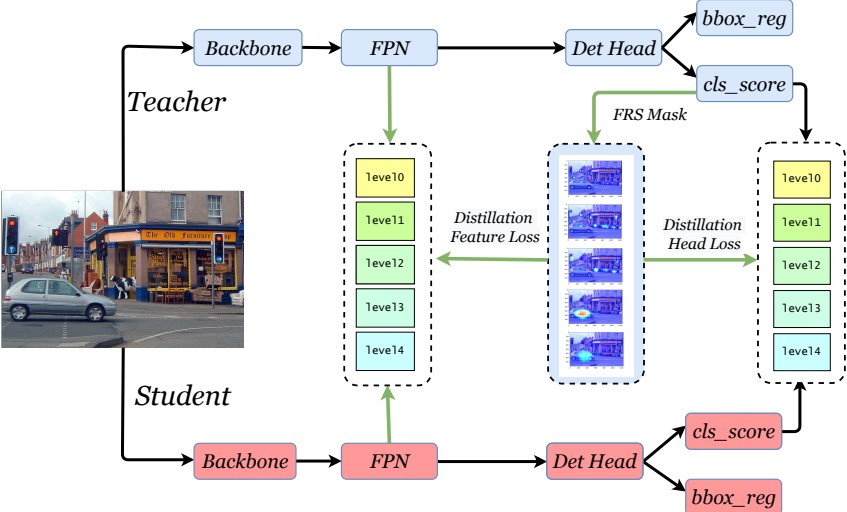

Figure 2: Overview of the proposed distillation via Feature Richness Score (FRS) framework. Each FPN layer generates a distillation mask which represents feature richness, and we use the corresponding mask on distilling both the FPN layers and classification head.

## 3 Distilling Object Detectors with Feature Richness Score

In previous works, most knowledge distillation methods pay more attention to the features in bounding box areas, while ignoring the important feature outside the bounding box and detrimental feature within the bounding box. To address this issue, we propose the Feature Richness Score as the criterion for selecting distillation areas. As shown in Figure 2, we use the aggregated classification score to represent the feature richness and then perform distillation on both FPN layers and the classification head.

### 3.1 Feature Richness Score Module

Given feature $f$, the probability that these features are objects can be described as:

$$y_c = P(c|f), \tag{1}$$

where $c$ represents an object category whether included in the dataset or not, so $y_c$ is the probability that feature $f$ is an object of category $c$. This conditional probability can be modeled with a deep neural network, which can be expressed as a parameter model $\theta$, described as:

$$y_c = P(c|f, \theta). \tag{2}$$

The proposed feature richness is defined as the probability that these features are objects, whether the categories of objects are included in the dataset or not. To make a summary of all categories, we can get the feature richness by aggregating the probability of all categories. Here we use maximum to perform the aggregating:

$$S = \max_c y_c = \max_c P(c|f, \theta). \tag{3}$$

In the framework of distillation, we use the trained classification head of teacher detector $\theta_t$ as an instantiation of $\theta$. At the same time, since there are objects of various categories, we use the categories $c' \in [1, C]$ in the dataset as the samples to represent all the objects $c$, where $C$ is the number of categories in the dataset. Therefore, the feature richness can be approximated as a classification probability of teacher detector:

$$S = \max_{1 \leq c' \leq C} y_{c'}^t = \max_{1 \leq c' \leq C} P(c'|f^t, \theta^t), \tag{4}$$

where $y^t$ is the classification probability of teacher detector, namely the classification score map which is the output of classification head from teacher detector, $f^t$ is the features fed into the classification head of teacher detector. In practice, we can get the feature richness $S$ by integrating the classification score map output of the teacher detector in the direction of the channel.

## 3.2 FPN layers

Nowadays, the FPN module is an important part of the object detection frameworks. Since FPN integrates multiple layers of the backbone, performing distillation on FPN can benefit both FPN itself and the backbone. FPN module is proposed to handle objects of various scales with multiple FPN layers. The shallow layers have a high resolution to detect objects of small scales, while deep layers have a low resolution to detect objects of large scales. Therefore, it is necessary that FPN layers should use separate feature richness masks which represent the probability of objects of different scales and are in different resolutions. We use the aggregated classification score map from the corresponding FPN layer as the feature richness:

$$S_l = \max_{1 \le c \le C} y_{lc}^t, \tag{5}$$

where $S_l$ is the feature richness mask of the $l$-th FPN layer, $y_l^t$ is the classification score map of the $l$-th FPN layer, C represents the number of channels. Then the distillation loss of FPN is:

$$L_{FPN} = \sum_{l=1}^{M} \frac{1}{N_l} \sum_{i=1}^{W} \sum_{j=1}^{H} S_{lij} \sum_{c=1}^{C} (F_{lijc}^t - \phi_{adapt}(F_{lijc}^s))^2, \tag{6}$$

where $M$ is the number of FPN layers, $l$ represents the $l$-th FPN layer, $i, j$ are the location of corresponding feature map with width $W$ and height $H$. $N_l = \sum_{i=1}^{W} \sum_{j=1}^{H} S_{lij}$. $F_l^t$ and $F_l^s$ are the $l$-th FPN features of student and teacher detectors, respectively. Function $\phi_{adapt}$ is the convolutional adaptation function to adapt $F_l^s$ to the same dimension as $F_l^t$.

## 3.3 Classification Head

For distilling the classification head, it is also necessary to use the proposed feature richness mask to effectively retrieve the important features outside the bounding boxes and remove the detrimental features within the bounding boxes. The feature richness mask used for classification head is the same as that of FPN in Eq.(5). The distillation loss of the classification head is described as:

$$L_{head} = \sum_{l=1}^{M} \frac{1}{N_l} \sum_{i=1}^{W} \sum_{j=1}^{H} S_{lij} \sum_{c=1}^{C} \phi(y_{lijc}^s, y_{lijc}^t), \tag{7}$$

where $l$ represents the $l$-th FPN layer, $y_l^t$ $y_l^s$ are the output of classification head of the $l$-th FPN layer from teacher model and student model respectively, $\phi$ refers to the Binary Cross-Entropy function used in classification head.

## 3.4 Overall loss function

The overall training targets of the student can be formulated as the weighted sum of the above-mentioned distillation losses alone with the conventional detection loss, described as:

$$L = L_{GT} + \alpha L_{FPN} + \beta L_{head} \tag{8}$$

where $L_{GT}$ is the detection training loss, $\alpha, \beta$ are hyper-parameters to balance different distillation losses in our method.

# 4 Experiments

In order to verify the effectiveness and robustness of our method, we conduct experiments on different detection frameworks and heterogeneous backbones, and on the COCO dataset. We choose the default 120k train images split for training and 5k val images split for the test. Meanwhile, we consider Average Precision as evaluation metric, i.e., mAP, $AP_{50}$, $AP_{75}$, $AP_S$, $AP_M$ $and$ $AP_L$. The last three measure performance with respect to objects with different scales.

Our implementation is based on mmdetection [4] with Pytorch framework. we use 2x learning schedule to train 24 epochs or the 1x learning schedule to train 12 epochs on COCO dataset. And the learning rate is divided by 10 at the 8-th and 11-th epochs for 1x schedule and the 16-th and 22-th epochs for 2x schedule. We set momentum as 0.9 and weight decay as 0.0001.

|  | mode | mAP | AP50 | AP75 | AP_S | AP_M | AP_L |
|---|---|---|---|---|---|---|---|
| Retina-Res101(teacher) | 2x | 38.9 | 58.0 | 41.5 | 21.0 | 42.8 | 52.4 |
| Retina-Res50(student) | 2x | 37.4 | 56.7 | 39.6 | 20.0 | 40.7 | 49.7 |
| ours | 2x | 39.7 | 58.6 | 42.4 | 21.8 | 43.5 | 52.4 |
| gain |  | +2.3 | +1.9 | +2.8 | +1.8 | +2.8 | +2.7 |
| GFL-Resnet101(teacher) | 2x | 44.9 | 63.1 | 49.0 | 28.0 | 49.1 | 57.2 |
| GFL-Resnet50(student) | 1x | 40.2 | 58.4 | 43.3 | 23.3 | 44.0 | 52.2 |
| ours | 1x | 43.6 | 61.9 | 47.5 | 25.9 | 47.7 | 56.4 |
| gain |  | +3.4 | +3.5 | +4.2 | +2.6 | +3.7 | +4.2 |
| GFL-Resnet101(teacher) | 2x | 44.9 | 63.1 | 49.0 | 28.0 | 49.1 | 57.2 |
| GFL-Resnet50(student) | 2x | 42.9 | 61.2 | 46.5 | 27.3 | 46.9 | 53.3 |
| ours | 2x | 44.7 | 63.0 | 48.4 | 28.7 | 49.0 | 56.7 |
| gains |  | +1.8 | +1.8 | +1.9 | +1.4 | +2.1 | +3.4 |
| Faster-Res101(teacher) | 1x | 39.4 | 60.1 | 43.1 | 22.4 | 43.7 | 51.1 |
| Faster-Res50(student) | 1x | 37.4 | 58.1 | 40.4 | 21.2 | 41.0 | 48.1 |
| ours | 1x | 39.5 | 60.1 | 43.3 | 22.3 | 43.6 | 51.7 |
| gains |  | +2.1 | +2.0 | +2.9 | +1.1 | +2.6 | +3.6 |
| FCOS-Resnet101(teacher) | 2x | 40.8 | 60.0 | 44.0 | 24.2 | 44.3 | 52.4 |
| FCOS-Resnet50(student) | 2x | 38.5 | 57.7 | 41.0 | 21.9 | 42.8 | 48.6 |
| ours | 2x | 40.9 | 60.3 | 43.6 | 25.7 | 45.2 | 51.2 |
| gains |  | +2.4 | +2.6 | +2.6 | +3.8 | +2.4 | +2.6 |

Table 1: Results of the proposed method with different detection frameworks. we use 2x learning schedule to train 24 epochs or the 1x learning schedule to train 12 epochs on COCO dataset.

| model | mode | mAP | AP50 | AP75 | APS | APM | APL |
|---|---|---|---|---|---|---|---|
| Retina-ResX101(teacher) | 2x | 40.8 | 60.5 | 43.7 | 22.9 | 44.5 | 54.6 |
| Retina-Res50(student) | 2x | 37.4 | 56.7 | 39.6 | 20 | 40.7 | 49.7 |
| [10] | 2x | 37.8 | 58.3 | 41.1 | 21.6 | 41.2 | 48.3 |
| [32] | 2x | 39.6 | 58.8 | 42.1 | 22.7 | 43.3 | 52.5 |
| ours | 2x | **40.1** | **59.5** | **42.5** | 21.9 | **43.7** | **54.3** |
| Retina-Res101(teacher) | 2x | 38.1 | 58.3 | 40.9 | 21.2 | 42.3 | 51.1 |
| Retina-Res50(student) | 2x | 36.2 | 55.8 | 38.8 | 20.7 | 39.5 | 48.7 |
| Fitnet [25] | 2x | 37.4 | 57.1 | 40 | 20.8 | 40.8 | 50.9 |
| General Instance [5] | 2x | 39.1 | 59 | 42.3 | 22.8 | 43.1 | 52.3 |
| ours [1] | 2x | **39.3** | 58.8 | 42 | 21.5 | **43.3** | **52.6** |

Table 2: Comparison with previous work with different detection frameworks.

## 4.1 Different detection frameworks

We verify the effectiveness of our proposed FRS on multiple detection frameworks, including anchor-based one-stage detector RetinaNet [18] and GFL [17], anchor-free one-stage detector FCOS [27], and two-stage detection framework Faster-RCNN [24] on COCO dataset [19]. As shown in Table 1. ResNet50 is chosen as the student detector, ResNet101 is chosen as the teacher detector. The results clearly show that our method gets significant performance gains from the teacher and reaches a comparable or even better result to the teacher detectors. RetinaNet with ResNet50 achieves 39.7% in mAP, meanwhile surpasses the teacher detector by 0.8% mAP. The ResNet50 1x based GFL surpasses the baseline by 3.4% mAP. The ResNet50 based on FasterRCNN gain 2.4% mAP. FCOS with ResNet50 achieves 40.9%, which also exceeds the teacher detector. These results clearly indicate the effectiveness and generality of our method in both one-stage and two-stage detectors.

## 4.2 Comparison with State-of-the-arts

Table 2 shows the comparison of the results of state-of-the-art distillation methods on the COCO [19] benchmark, including bounding boxes based methods [28]. As shown in Table 2, the performance

---

[1]For fair comparison, we run the experiment using the teacher and student model of the same accuracy, the implementation is based on the mmdetection 1.2. In addition to this set of experiments, our other experiments are based on mmdetection2

| | | | | | |
|---|---|---|---|---|---|
| Retina-Res50 | ✓ | | ✓ | ✓ | ✓ |
| Retina-Res101 | | ✓ | | | |
| FPN layers | | | ✓ | | ✓ |
| Classification Head | | | | ✓ | ✓ |
| mAP | 37.4 | 38.9 | 39.4 | 38.4 | 39.7 |

Table 3: Ablation Study for various distillation modules on COCO dataset

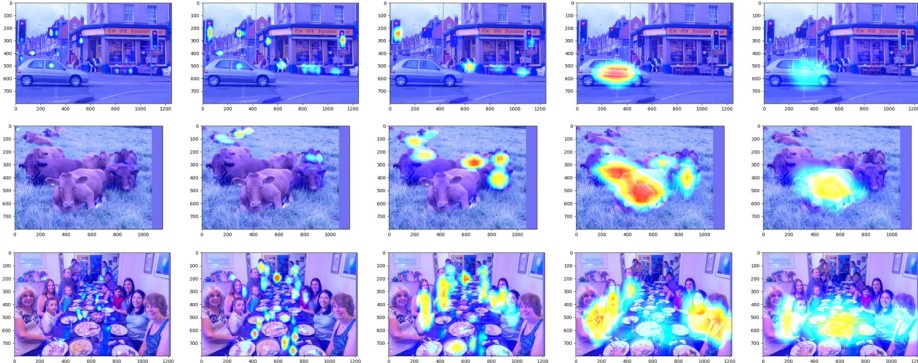

Figure 3: From left to right, they correspond to the feature masks of different FPN layers. The leftmost corresponds to the lowest level of FPN , which identifies small objects, and the rightmost corresponds to the highest level of FPN, which is used to identify large objects.

of the student detector has been greatly improved. For example, under the guidance of RseNext101 teacher detector, the ResNet50 with RetinaNet student detector achieves 40.1% in mAP, surpassing [32] by 0.5% mAP. With the help of ResNet101 teacher detector, the ResNet50 based RetinaNet gets 39.3% mAP, which also exceeds the State-of-the-art methods General Instance [5]. The results demonstrate that our method has achieved a good performance improvement, surpassing the previous SOTA methods.

## 4.3    Ablation Study

### 4.3.1    Impact of different modules

In this part, we conducted some ablation experiments with different distillation modules to better reflect the impact of each module on knowledge distillation. As shown in Table 3, each distillation module can improve the performance of the student network. Distillation on the FPN module can bring an improvement of 2.0% mAP, and distillation on the classification head module can gain 1.0% mAP. The combination of both can further improve performance, which brings up to 2.3% mAP improvement. It can be seen that distillation on each module has a certain improvement for knowledge distillation.

### 4.3.2    Varying teacher detector for FRS

In this subsection, we investigate the influence of different teacher detectors for distillation. The student detector is RetinaNet (whose backbone is RegNet). As shown in Figure 6, it is obvious that with the improvement of the accuracy of the teacher's network, the performance of the students has also been improved.

## 4.4    Analysis

### 4.4.1    Qualitative performance gain from imitation

We qualitatively compare the results of student detector and our FRS. In Figure 4, the upper row images show the results of student detector without distillation, while the lower row shows the

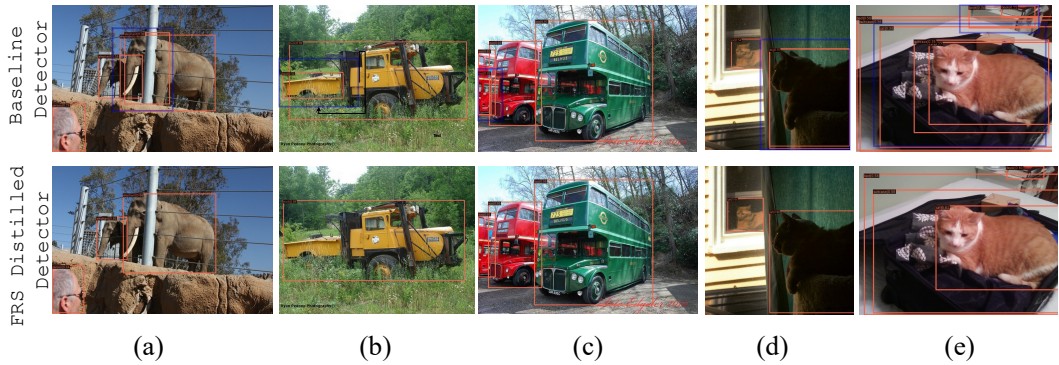

|     | (a) | (b) | (c) | (d) | (e) |
| --- | --- | --- | --- | --- | --- |

Figure 4: Qualitative analysis on COCO2017 dataset with distilled and baseline Retianet-ResNet50. The blue boxes mean the undetected or wrongly detected objects of the baseline detector

distillation results of the FRS method. Obviously, the detectability of the student network has been improved by the FRS distillation method. As shown in Figure 4(a)(c), The student detector generates multiple bounding boxes for the same object which are unfortunately not able to be suppressed by NMS. while our method can accurately locate the location of the same objects that are close to each other, avoiding generating incorrect bounding box. Figure 4(b) shows the student model mistakenly detects a boat, while our method avoids this error and outputs a rather accurate bounding box for the trunk instance. As shown in Figure 4(d), the raw student model mistakes the cat for a dog. In Figure 4(e), the student model wrongly predicts an area of background as a cat instance, while the imitated student avoids the error, indicating that the FRS method has a better ability to discriminate.

### 4.4.2 Visualization of FRS

As shown in Figure 3, we visualize the heat map from each FPN level by the FRS method to better understand the importance of different regions. It can be clearly seen that the region extracted by FRS obviously covers the objects of the corresponding scale. As for the edge area of objects, the corresponding feature weight is smaller, while in the regions with rich feature information, such as the central area of the object, the corresponding feature weight is relatively high. Meanwhile, the highlight areas of different FPN levels are different and have different semantic information. The low-level FPN is more focused on finding the characteristics of small objects, while the high-level FPN level pays more attention to large objects. Besides, the distance also affects the visual size. The FRS method can better extract subtle information and find the most suitable one from different FPN levels.

FRS method is pixel-wise and each pixel is in a form of soft-label which can carry more information than the hard-label. In foreground areas, as shown in Figure 3, the closer to the center of the object, the richer the features and the higher the corresponding value. In the edge area of the object, the detector can only observe the local features of the object, which leads to a lower feature richness score.

### 4.4.3 Comparison of different distillation areas

To further analyze the contribution of FRS methods, we investigate the influence of the different regions. In order to compare the difference between the FRS method and the previous ground truth method in detail, we divide the image area into four parts according to the feature extraction situation of the teacher network, namely TP(True Positive), FP(False Positive), FN(False Negative), TN(True Negative). As shown in Table 4, distilling the TP+FP regions performs the highest mAP, also higher than using TP+FN regions. The area selected by the previous ground truth method corresponds to TP+FN regions, while the FRS method corresponds to TP+FP regions, since it thinks the FP regions are as important as TP regions but ignores the FN regions. These results show that the proposed FRS outperforms than existing methods of distilling based on bounding boxes in a different aspect. Moreover, the FRS method which focuses on TP+FP regions is better than distilling only the TP area, which proves that the FP region is effective for distillation. Besides, distilling on TP+FN regions shows lower than that on only TP regions, which shows distilling on FN may damage the results.

|       | mAP  | AP50 | AP75 | AP_S | AP_M | AP_L |
|-------|------|------|------|------|------|------|
| TP    | 39.1 | 58.2 | 41.8 | 21.3 | 43.0 | 51.7 |
| TP+FP | 39.4 | 58.4 | 42.1 | 21.6 | 43.7 | 52.1 |
| TP+FN | 39.0 | 58.0 | 41.9 | 21.5 | 43.1 | 51.9 |
| TP+TN | 38.7 | 57.8 | 41.2 | 20.6 | 42.3 | 52.3 |

Table 4: Based on the ground truth and classification score of the teacher detector, we have made a more detailed division of the foreground and background area. TP (True Positive) refers to the foreground area with high feature richness scores, FN (False Negative) means the foreground area with low feature richness scores, FP (False Positive) is the background area with high scores, and TN (True Negative) denotes the background area with low scores. For clarity, we only distill the FPN layers on Retianet-ResNet50 model, under the guidance of ResNet101 detector

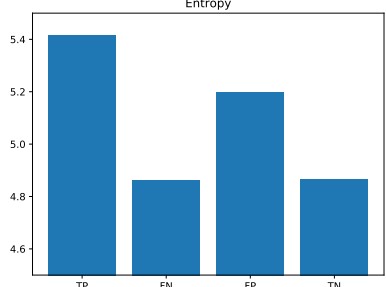

Figure 5: Information entropy of different regions.

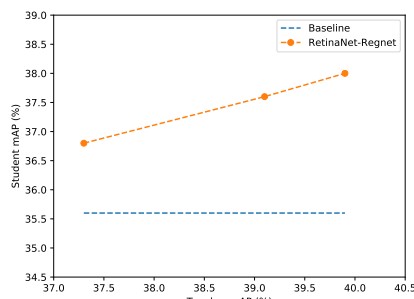

Figure 6: Relation between teacher and student detector on Regnet based Retinanet in COCO2017 dataset

Meanwhile, we also compared the distillation on the TP+TN regions, and the result is no better than that on only the TP area, which shows that TN has no effect on distillation.

We analyzed the FRS feature extraction method from the perspective of information entropy. We find that the FRS method can extract more feature-rich areas from the background area, which are the areas with higher information entropy. As shown in Table 5, the information entropy of the TP area is the highest, and the information entropy corresponding to the FP area is also relatively high. The rest of TN and FN have the lowest information entropy. Therefore, the area with higher information entropy corresponds to more feature information, which can promote the distillation effect.

# 5 Conclusion

In this paper, we propose an effective distillation method for object detection. We analyze and demonstrate the beneficial features outside the bounding box and the detrimental features within the bounding boxes. Based on this finding, we propose the FRS method to distill important features that can improve the detectability of the student detector. FRS method effectively improves the performance of modern detection frameworks and can be universally applicable for both two-stage and one-stage detection frameworks.

# Acknowledgments

This work is partially supported by the Beijing Natural Science Foundation (JQ18013), the NSF of China(under Grants 61925208, 61906179, 62002338, 61732007, 61732002,62102399), Strategic Priority Research Program of Chinese Academy of Science (XDB32050200), Beijing Academy of Artificial Intelligence (BAAI), CAS Project for Young Scientists in Basic Research(YSBR-029) and Xplore Prize.

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
