# 1 Experiments on more models and datasets

## 1.1 More dataset

Pascal VOC dataset is widely used in object detection, so we conduct the extend experiment results on Faster R-CNN and RetianNet on Pascal VOC. The ResNet50 based FasterRCNN surpasses the baseline by 1.7% mAP. The ResNet50 based on RetinaNet gain 2.9% mAP. It can be seen that our method has also achieved significant improvements on other data sets.

| Model | AP50 |
|---|---|
| faster-Res101(teacher) | 81.3 |
| faster-Res50(student) | 79.5 |
| ours | 81.2 |
| gains | 1.7 |
| Retina-Res101(teacher) | 80.1 |
| Retina-Res50(student) | 77.3 |
| ours | 80.2 |
| gains | 2.9 |

## 1.2 More detector

YOLOF is the latest detector without FPN module, so it is different in construction from the detector in our paper. Therefore YOLOF is suitable to evaluate the generalization of our methods. The following table shows the experiment results on YOLOF on MS COCO2017. YOLOF with ResNet50 achieves 39.3% in mAP, meanwhile surpasses the baseline by 1.8% mAP.

| model | mAP | AP50 | AP75 | AP_S | AP_M | AP_L |
|---|---|---|---|---|---|---|
| YOLOF-Res50(student) | 37.5 | 57 | 40.4 | 19 | 42 | 53.2 |
| YOLOF-Res101(teacher) | 39.7 | 59 | 42.9 | 20.5 | 44.3 | 55 |
| ours | 39.3 | 58.5 | 42.5 | 20.5 | 44 | 54.5 |

# 2 Performance gain from object regions

Figure 1 is an analysis of the PR curves of different sizes of target objects. We compare the student network, the bounding box based distillation method, and the FRS method. We find that the FRS method has a relatively good improvement for Cor, Loc, Sim, and Oth. Besides, it improves the big objects especially obviously. The result demonstrates that the FRS method can well promote the feature extraction ability of student detector learning teacher detector.


Figure 1: Precision-Recall curves of different distillation methods on COCO benchmark. The student detector is RetinaNet50, and The teacher detector is RetinaNet101