# OpenReview forum: "Distilling Object Detectors with Feature Richness"
_NeurIPS.cc/2021/Conference — NeurIPS 2021 Poster_

### Official Review · Reviewer_KFUi · 2021-07-15

**Rating:** 6
**Confidence:** 5

**Summary:**

In this paper, the authors proposed feature-richness score to choose important features for knowledge distillation. Specifically, the proposed method uses the classification score from teacher model as the feature-richness masks. The feature-richness masks are used as indicators that show where the import feature is and what the importance of this feature. Based on these masks, the proposed method can focus on distilling import features in FPN and classification head.

**Limitations And Societal Impact:**

The authors have adequately addressed the limitations and potential negative societal impact of their work.

**Main Review:**

Strengths
- The proposed method is simple but effective. In the detectors with FPN, the improvements is great.
- This paper is well written and readable.

Weaknesses
- In the experiments, the authors valid the effectiveness of the proposed method in detectors with FPN. But there are also many detectors without FPN, such as [1] and [2]. Could you show the improvements in these detectors with the proposed method?
- This paper should compare with other recent state-of-the-art methods, e.g. [3].
- The distillation losses in the proposed method are related to two hyper-parameters: $\alpha$ and $\beta$. How these hyper-parameters affect the performance? Does the values of these hyper-parameters depend on datasets or detectors?
- For Table 3, the performance Retina-Res50 is 38.5 mAP while  Retina-Res101 has 37.4 mAP. Is it fault? Besides, even if these two values exchange, for Retina-Res101, 38.5 mAP in Table 3 is different from 38.9 mAP in Table 1. In addition, in line 203 and line 204, the mAP improvements (1.9% and 1.1%) in text are different from ones, as shown in Table 3. Can you explain these mismatches?
- In equation (5) and equation (6), the $c'$ and $C$ should denote different things, one for classes and the other for channels. It need to be clarified in the paper.
- The proposed method is only valid on COCO dataset, while other methods, such as General Instance[5] and Fine-grained[27], also conduct experiments on PASCAL VOC. Could you show the results on PASCAL VOC with the proposed method?

[1] Duan, Kaiwen, et al. "Centernet: Keypoint triplets for object detection."Proceedings of the IEEE/CVF International Conference on Computer Vision. 2019.\
[2] Chen, Qiang, et al. "You only look one-level feature."Proceedings of the IEEE/CVF Conference on Computer Vision and Pattern Recognition. 2021. \
[3] Zhang, Linfeng, and Kaisheng Ma. "Improve Object Detection with Feature-based Knowledge Distillation: Towards Accurate and Efficient Detectors."International Conference on Learning Representations. 2021.

**Time Spent Reviewing:**

24

---

> ### Author Response · Authors · 2021-08-10
> **Response**
>
> We would appreciate your advice for the further improvement of our work. We would like to address your concerns one by one. Any further discussion would be appreciated.
>
> **Question1. About the detector without FPN**
>
> Considering the Centernet detector requires too long training time and we need to train both the teacher detector and the distilled student network, and the time is far from enough during the rebuttal period. Therefore, we only conduct experiments on YOLOF detector. The following table shows the experiment results on YOLOF on MS COCO2017. YOLOF with ResNet50 achieves 39.3% in mAP, meanwhile surpasses the baseline by 1.8% mAP.
>
> | Model                 | AP   | AP50 | AP75 | APS  | APM  | APL  |
> |-----------------------|------|------|------|------|------|------|
> | YOLOF-Res50(student)  | 37.5 | 57   | 40.4 | 19   | 42   | 53.2 |
> | YOLOF-Res101(teacher) | 39.7 | 59   | 42.9 | 20.5 | 44.3 | 55   |
> | ours                  | 39.3 | 58.5 | 42.5 | 20.5 | 44   | 54.5 |
>
> **Question2. Comparison with SOTA method**
>
> we conduct experiments on RetianNet on MS COCO2017 and compare with [3].
>
> **Results** The results are as follows: Our method surpasses [3] by 0.5% mAP. It can be seen that our method has achieved a better improvement effect.
>
> |                         | mAP  | AP50 | AP75 | APS  | APM  | APL  |
> |-------------------------|------|------|------|------|------|------|
> | Retina-Res50(student)   | 37.4 | 56.7 | 39.6 | 20   | 40.7 | 49.7 |
> | Retina-ResX101(teacher) | 40.8 | 60.5 | 43.7 | 22.9 | 44.5 | 54.6 |
> | [3]                     | 39.6 | 58.8 | 42.1 | 22.7 | 43.3 | 52.5 |
> | ours                    | 40.1 | 59.5 | 42.5 | 21.9 | 43.7 | 54.3 |
>
> **Question3. About hyper-parameters**
>
> We believe that the hyperparameters are affected by the detector, and different hyperparameters are used in the two stages and the first stage. Different data sets require different hyperparameters. But compared with other methods [3] [4], we use very few hyperparameters.
> In the initialization hyperparameters, we refer to the fine-grained[5] paper. The initial setting of the hyperparameters keeps the distillation loss and the distillation loss in this paper at an order of magnitude.
>
> **Question4. About Table 3**
>
> Thank you for helping to correct the wrong place, we will correct it immediately.
>
> | Model               |      |      |      |      |      |
> |---------------------|------|------|------|------|------|
> | Retina-Res50        | √    |      | √    | √    | √    |
> | Retina-Res101       |      | √    |      |      |      |
> | FPN layers          |      |      | √    |      | √    |
> | Classification Head |      |      |      | √    | √    |
> | mAP                 | 37.4 | 38.9 | 39.4 | 38.4 | 39.7 |
>
> "Distillation on the FPN module can bring an improvement of 2.0% mAP, and distillation on the classification head module can gain 1.0% mAP. "
>
> **Question5. About the equation**
>
> Thanks for the suggestion, we will declare it in more detail in the paper.
>
> **Question6. Experiments on more datasets**
>
> We conduct the extend experiment results on Faster R-CNN and RetianNet on Pascal VOC.
> The ResNet50 based FasterRCNN surpasses the baseline by 1.7% mAP. The ResNet50 based on RetinaNet gain 2.9% mAP. It can be seen that our method has also achieved significant improvements on other data sets.
>
> | Model                  | AP50 |
> |------------------------|------|
> | faster-Res101(teacher) | 81.3 |
> | faster-Res50(student)  | 79.5 |
> | ours:                  | 81.2 |
> | gains:                 | 1.7  |
> | Retina-Res101(teacher) | 80.1 |
> | Retina-Res50(student)  | 77.3 |
> | ours                   | 80.2 |
> | gains                  | 2.9  |
>
> [1] Duan, Kaiwen, et al. "Centernet: Keypoint triplets for object detection."Proceedings of the IEEE/CVF International Conference on Computer Vision. 2019.
>
> [2] Chen, Qiang, et al. "You only look one-level feature."Proceedings of the IEEE/CVF Conference on Computer Vision and Pattern Recognition. 2021.
>
> [3] Zhang, Linfeng, and Kaisheng Ma. "Improve Object Detection with Feature-based Knowledge Distillation: Towards Accurate and Efficient Detectors."International Conference on Learning Representations. 2021.
>
> [4]General instance distillation for object detection, CVPR2021
>
> [5]Distilling Object Detectors with Fine-grained Feature Imitation,CVPR2019

---

> > ### Comment · Reviewer_KFUi · 2021-08-30
> > **Update**
> >
> > I have read all the reviews and author's feedback. The author's feedback has resolved my concerns, thus I increase my rating from 5 to 6.

---

### Official Review · Reviewer_uYvK · 2021-07-15

**Rating:** 7
**Confidence:** 5

**Summary:**

This paper introduces a novel knowledge distillation method for object detection, named Feature-Richness Score(FRS), which makes good use of aggregated classification score of the feature map as the mask of FRS. This pixel-wise FRS mask would be used as the weighted matrix to guide the distillation from teacher to student within selected layers, such as FPN and head. Plenty of well-designed experiments could show the effectiveness and availability of their method on serval mainstream detection framework, also prove it surpasses other recent methods.

**Ethical Concerns:**

Don't have any Ethical Concerns with this submission.

**Limitations And Societal Impact:**

1. Typo: In Figure 2, the second level 1 in graph.
2. Correction might be needed in Table 2.
3. The readability of Section 4.3.1 and Table 3 could be improved.

**Main Review:**

Pros:
1. The proposed method is quite clean and simple, but effective, compared with other recent methods with complicated design. It just use the aggregated classification score, take maximum operation in channel direction, of the feature map as a weighted score mask, named Feature-Richness Score(FRS) to guide the distillation, which is easy to implement and plug into any layers.
2. Unlike the prior methods, the proposed method could make use of whole feature map, not only features around bbox area, which most methods usually do. That makes the authors claim, avoiding the defected results from teacher model, sounds reasonable and understandable.
3. The authors had done extensive experiments on serval mainstream detection frameworks.

Cons:
1. In Table 2, the accuracy of teacher model (Retina-res101, mAP=38.9) is higher than the accuracy (mAP=38.1) of the same model used by General Instance method, but the compared accuracy of General Instance seems like to be just copied from their work, which makes the comparison is not fair. Same issue on the results of faster-RCNN framework.
2. In Table 4, the improvement (+0.3) from TP to TP+FP may prove the potential gain from make use of out of bbox features is limited, combining with Cons#1, the gap between the proposed method and previous sota might be shorten.
3. The FRS design is highly related to the category size. what if the dataset has limited classes, which is a usual case in autonomous drive scenario. Think about an extreme case with just one class, it seems it would fall back to some kind of trivial baseline.
4. The Table 3 is quite confusing, why the Res50 surpasses Res101? Also, the results of the last three columns didn't mention the backbone they used.

Originality: The motivation of the proposed method differs from previous contributions. And the design is simple and clear.

Quality: The submission has extensive experiments to support their claims, but the results presentation could be improved.

Clarity: The submission is easy to read and understand.

Significance: The submission would be a meaningful contribution to this research domain and bring a new point of view to make use of features out of bbox.

**Time Spent Reviewing:**

3~4

---

> ### Author Response · Authors · 2021-08-10
> **Response**
>
>
> We are encouraged that you think our work is of great quality and good significance. We would also appreciate your constructive suggestions for further improvement of our work. We would like to discuss these suggestions one by one. Any further discussion will be appreciated.
>
> **Question1. The inconsistent results of teacher and student detectors**
>
> We found that the implementation accuracy in this GI paper is based on the mmdetection 1.2 framework, so we re-run our code on the mmdetection 1.2 framework and keep it consistent with the accuracy of the teacher network in [5]. The results are as follows: Our method surpasses [5] by 0.2% mAP. It can be seen that our method has achieved a better improvement effect.
>
> |                        | mAP  | AP50 | AP75 | APS  | APM  | APL  |
> |------------------------|------|------|------|------|------|------|
> | Retina-Res50(student)  | 36.2 | 55.8 | 38.8 | 20.7 | 39.5 | 48.7 |
> | Retina-Res101(teacher) | 38.1 | 58.3 | 40.9 | 21.2 | 42.3 | 51.1 |
> | [5]                    | 39.1 | 59   | 42.3 | 22.8 | 43.1 | 52.3 |
> | ours                   | 39.3 | 58.8 | 42   | 21.5 | 43.3 | 52.6 |
>
> **Question2 The different with TP and sota**
>
> TP is not an existing method, but in order to prove the effectiveness of FP, we removed the FP area in our original experiment and then performed the experiment. The student detector, teacher detector and hyperparameter settings used in this part of the experiment are exactly the same.
>
> **Question3 About the category size**
>
> In extreme cases, even if there is only one category, this category can be used to represent categories outside the data set that are similar to it, so you can still have more information from categories outside the data set. Therefore, our method will still achieve a better improvement than the baseline.
>
> We also conduct experiments on Pascal VOC dataset with a smaller category size.
> The ResNet50 based FasterRCNN surpasses the baseline by 1.7% mAP. The ResNet50 based on RetinaNet gain 2.9% mAP. It can be seen that our method has also achieved significant improvements on the dataset with a smaller category size.
>
> | Model                  | AP50 |
> |------------------------|------|
> | faster-Res101(teacher) | 81.3 |
> | faster-Res50(student)  | 79.5 |
> | ours:                  | 81.2 |
> | gains:                 | 1.7  |
> | Retina-Res101(teacher) | 80.1 |
> | Retina-Res50(student)  | 77.3 |
> | ours                   | 80.2 |
> | gains                  | 2.9  |
>
> **Question4 About the Table 3**
>
> Thank you for helping to correct the wrong place, we will correct it immediately.
>
> | Model               |      |      |      |      |      |
> |---------------------|------|------|------|------|------|
> | Retina-Res50        | √    |      | √    | √    | √    |
> | Retina-Res101       |      | √    |      |      |      |
> | FPN layers          |      |      | √    |      | √    |
> | Classification Head |      |      |      | √    | √    |
> | mAP                 | 37.4 | 38.9 | 39.4 | 38.4 | 39.7 |

---

> > ### Comment · Reviewer_uYvK · 2021-09-02
> > **Update**
> >
> > Thanks for the detailed response from authors. I tend to keep my rating of score 7.

---

### Official Review · Reviewer_iJCP · 2021-07-16

**Rating:** 6
**Confidence:** 4

**Summary:**

One of the challenges in object detection is to train models with reduced computational and memory complexity for deployment on the resource-limited devices. One of the central methods is knowledge distillation which transfers knowledge from large teacher detector into small one with almost preserving the quality of the teacher model. However, existing methods have several issues: i) they ignore feature representation outside bounding boxes which can be helpful (e.g. similar but absent from the class categories objects which detection can improve target class detection)  ii) they use some detrimental representations within bounding boxes (ignore teacher model mistakes). To overcome these issues the paper proposes a novel method (FRS: Feature-Richness Score) to choose important features which can improve detection during distillation process. With experiments on anchor-based/anchor-free and one-stage/two-stage detectors authors demonstrate significant improvement in detection with FRS: FRS outperformes state-of-the-art methods on all distillation-based detection frameworks, Faster-RCNN, Retinanet, GFL and FCOS.

**Limitations And Societal Impact:**

Authors did not specify any limitations or social impact throughout the text and in the supplementary, however in the checklist it is marked as "Yes". From my point I don't see any potential problem for societal impact except that detection can be further improved and additional bias with new training method should be later investigated. As limitations - obviously it will worse work with detectors without FPN layers as we cannot use extra info to define important regions.

**Main Review:**

**Originality**

Knowledge distillation is a well known approach and widely used for different tasks including object detection. Specifically for object detection task imitating all features is inefficient because there is a lot of object-irrelevant information. So the main focus of research for knowledge distillation in object detection is how to select proper representations to distill. Instead of following common approach to select features from the ground truth bounding boxes, in this paper a novel approach how to select/define important features for distillation is proposed: features importance can be seen as probability that these features are objects (pixel-wise aggregated classification score is used as this importance measure). Proposed method generates pixel-wise importance masks for each FPN layer and applicable to any detector type: one-stage, two-stage, anchor-free. Section 2 well covers related works and comparison between existed methods and the one proposed in the paper.

**Quality**

There is no any theoretical analysis and work is purely experimental. Experiments are performed
- across different detectors with standard distillation approach; proposed method demonstrates significant gains and even sometimes outperforms the teacher (Table 1)
- across different SOTA distillation approaches for different detectors (Table 2) - here some clarification on the Table is needed (see comment below)
- for ablations on each distillation loss component (Table 3)
- for ablation on varying the teacher detector performance (better teacher -> better student)
- for detailed analysis of the proposed method, demonstrating that indeed usage of the most informative parts (TP and FP) improves distillation.

The following additional information on experiments is needed to conclude if comparison is technically correct:
- What will happen if learning schedule will be longer for student in table 1?
- Table 2: It seems results for all methods except teacher and student are taken from [5], and in [5] teacher and student baselines are reported to be worse than ones in the paper. Did authors retrain teacher and student on their own? Did they retrain all other state-of-the-art methods? For example in [5] teacher is reported 38.1 (-0.8 from the number in Table 2) which can be the source of inconsistent results in Table 2 due to better teacher used for the paper's method (could be that [5] outperforms proposed method with the same teacher). This place should be clarified and fixed in the text.

Other comments/questions:
- YOLO is also one of the leading detectors, what about it?
- line 133-135: sentence is not clear. Could author explain what they mean?
- It is not clear from the text how categories/classes out of training dataset can be used, how the feature richness score is computed in this case?
- Eq. (6) why is $\phi_{adapt}$ applied to the student, not to the teacher? I assume teacher has larger dimension, so it is more reliable to perform downsampling rather than upsampling.
- line 203: 1.9% should be 2.0% looking at Table 3; line 204: 1.1% should be 1.0% looking at Table 3.
- Table 3 is possible incorrect for first two columns. In table 1 Retine-Res101 is reported as 38.9 (37.4 in Table 3) and Retina-Res50 37.4 (38.5 in Table 3).

**Clarity**

Paper is mostly clear written, however there are some misleading sentences/formulations, a lot of typos and probably it is worth to have some native-speaker proof-reading for the text to improve readability. Below the list (not the full one) of typos and additional comments/suggestions on improving the text:
- line 55: "the importance of features is strongly correlated with the feature richness". I would say importance of features by paper's definition is feature richness. Probably this sentence should be removed or reformulated.
- Please introduce FPN notation before its usage
- line 68: "Finally, Our" -> "Finally, our"
- line 71: " FRS, We" -> " FRS, we", "on various framework" -> "on various framework**s**"
- throughout the text fix listing several references together, for example line 78 "[23]  [2][9]" -> "[23,2,9]"
- line 79-80: reformulate sentence as the verb is absent in it.
- line 82: fix is needed in "applications. while"
- line 87-88: fix English in the sentence - either remove first "while" or put some verb in the second sentence part.
- line 95: "Hinton [11] proposal" -> "Hinton [11] proposed"
- line 96: "which open" -> "which open**s**"
- line 99: "a important role" -> "an important role"
- line 101: "The relationship" -> "the relationship"
- line 104: "[3] proposal to distills" -> "[3] proposed to distill"
- be consistent with "et al"/ "et al." usage
- lines 107, 109: two spaces before references
- line 143: "perform distillation" -> "performing distillation"
- line 154: "student and teacher detector" -> "teacher and student detectors"
- line 172-173 ", We" -> ", we"; fix "and" in the formula
- line 175: ". we" -> ". We"
- Template guidelines: tables captions should be at the top of the tables not at the bottom
- line 191: "The table 2" -> "Table 2"
- Table 2: Fater-Res101 -> Faster-RCNN? R-CNN?
- line 204 "can further improved" -> "can further improve"
- Figure 5 - text size is different from other
- line 214: "In Figure 4, The" -> "In Figure 4, the"
- line 219: ". while Our" -> ", while our"
- line 220: "bbox" -> "bounding box" or introduce the abbreviation

Necessary details on the training configuration are provided, so should be reproducible.

**Significance**

Results on distillation are important as we can further improve and simplify the student network training for deployment purposes.  In this paper authors propose a novel method how to define important regions for distillation and demonstrated with in-depth analysis its effectiveness. Proposed method is simple in its implementation, applicable to different detectors and outperforms state-of-the-art-models (should be rechecked with Table 2 + more info from authors).

**Update**: As authors addressed to all my concerns and fixed the Table 2, I change the final decision from 5 to 6

**Time Spent Reviewing:**

12

---

> ### Author Response · Authors · 2021-08-10
> **Response**
>
> We would appreciate your advice for the further improvement of our work. We would like to address your concerns one by one. Any further discussion would be appreciated.
>
> **Question1. The inconsistent results of teacher and student detectors**
>
> We found that the implementation accuracy in this GI paper is based on the mmdetection 1.2 framework, so we re-run our code on the mmdetection 1.2 framework and keep it consistent with the accuracy of the teacher network in [5]. The results are as follows: Our method surpasses [5] by 0.2% mAP. It can be seen that our method has achieved a better improvement effect.
>
> |                        | mAP  | AP50 | AP75 | APS  | APM  | APL  |
> |------------------------|------|------|------|------|------|------|
> | Retina-Res50(student)  | 36.2 | 55.8 | 38.8 | 20.7 | 39.5 | 48.7 |
> | Retina-Res101(teacher) | 38.1 | 58.3 | 40.9 | 21.2 | 42.3 | 51.1 |
> | [5]                    | 39.1 | 59   | 42.3 | 22.8 | 43.1 | 52.3 |
> | ours                   | 39.3 | 58.8 | 42   | 21.5 | 43.3 | 52.6 |
>
> **Question2. the longer training**
>
> We conduct experiment on the Retina-Res50 as the student and the Retina-Res101 as the teacher detector. We use 3x learning schedule to train 36 epochs, 2x learning schedule to train 24 epochs, or the 1x learning schedule to train 12 epochs on COCO dataset. It can be seen that with the increase of time, the accuracy of training has also improved, but from 2x to 3x, the improvement is significantly smaller (0.2%). Therefore, we believe that the benefits of 3x are relatively small. In the paper, we think that 2x is enough.
>
> |    | AP   | AP50 | AP75 | APS  | APM  | APL  |
> |----|------|------|------|------|------|------|
> | 1x | 39   | 58.1 | 41.8 | 21.8 | 42.8 | 52   |
> | 2x | 39.7 | 58.6 | 42.4 | 21.8 | 43.5 | 52.4 |
> | 3x | 39.9 | 58.7 | 42.8 | 22.1 | 43.9 | 53.1 |
>
> **Question3. About YOLO**
>
> Considering the YOLO detector requires too long training time and we need to train both the teacher detector and the distilled student network, and the time is far from enough during rebuttal period (at least 10 days). Therefore, we use YOLOF detector instead. YOLOF is the latest detector without FPN module, so it is different in construction from the detector in our paper. Therefore YOLOF is suitable to evaluate the generalization of our methods. The following table shows the experiment results on YOLOF on MS COCO2017. YOLOF with ResNet50 achieves 39.3% in mAP, meanwhile surpasses the baseline by 1.8% mAP.
>
> | Model                 | AP   | AP50 | AP75 | APS  | APM  | APL  |
> |-----------------------|------|------|------|------|------|------|
> | YOLOF-Res50(student)  | 37.5 | 57   | 40.4 | 19   | 42   | 53.2 |
> | YOLOF-Res101(teacher) | 39.7 | 59   | 42.9 | 20.5 | 44.3 | 55   |
> | ours                  | 39.3 | 58.5 | 42.5 | 20.5 | 44   | 54.5 |
>
> **Question4. The unclear sentence**
>
> In the paper, c represents an object category in the broad sense, whether included in the dataset or not. That is to say, c represents any object category in the real world, while c' represents an object category in the dataset.  We think object categories in the dataset are the *samples* from the real world, so we can use c' as the *sample* to represent c. Here we introduce c' and c to explain the computing of feature richness. Feature richness is defined as the probability that features are objects in the real world. However, we are incapable to compute feature richness from the probability of all objects in the real world directly, so we compute it on the *samples* of the objects in the real world, i.e. the object categories in the dataset, as shown in Eqn3 and Eqn4.
>
> **Question5. How categories/classes out of training dataset can be used**
>
> The probability of a category not included in the dataset can be represented by the probability of a visually similar category included in the dataset. For example, the COCO dataset does not include mannequins category, but includes human category. Mannequins are visually similar to people. Therefore, when a mannequins is detected, the human category will have a higher probability score. The probability of the human category can be used to approximate the probability of the mannequins. Therefore, we believe that the FP area (the area with higher object probability outside the bounding box) can be used to represent the categories that are not included in the dataset. Our method has a large feature richness in the FP region, and can capture the information of the categories not included in the dataset contained in the FP region. Using the probabilities of categories not included in the dataset is helpful to improve the generalization of the detector. Experiments in Table 4 have also shown that using the FP area is effective for improving the distillation results.
>
>
> **Question6. About the adaptation**
>
> If the teacher network uses an adaptive layer, this will lead to collapse solutions.
> That means the output of the teacher network through the adaptive layer and the output of the student network are both zeros.
>
>
> **Question7. The wrong data**
>
> Thank you for helping to correct the wrong place, we will correct it immediately.
>
> | Model               |      |      |      |      |      |
> |---------------------|------|------|------|------|------|
> | Retina-Res50        | √    |      | √    | √    | √    |
> | Retina-Res101       |      | √    |      |      |      |
> | FPN layers          |      |      | √    |      | √    |
> | Classification Head |      |      |      | √    | √    |
> | mAP                 | 37.4 | 38.9 | 39.4 | 38.4 | 39.7 |
>
> "Distillation on the FPN module can bring an improvement of 2.0% mAP, and distillation on the classification head module can gain 1.0% mAP. "

---

> > ### Comment · Reviewer_iJCP · 2021-08-26
> > **Update**
> >
> > Thanks authors for detailed clarification, fixes and additional ablations. Several last comments:
> >
> > > Question1. The inconsistent results of teacher and student detectors
> >
> > Did you exactly reproduced now the baseline student and teacher with 1.2 framework version? I see now that improvements over [5] are small and even worse on AP50, AP75 and APS. Could it be that it is now just stat error?
> >
> > > Question2. the longer training
> >
> > About longer training I mean the baseline student model.
> >
> > > Question4. The unclear sentence
> >
> > For the final revision I would recommend to be more specific and clear in the text.
> >
> >
> >
> > Overall results looks convincing across different detectors, datasets and other existing methods. As authors provided the fixed Table 2 for which I had the main concern I raise the final rating from 5 to 6.

---

> > > ### Author Response · Authors · 2021-09-02
> > > **Response**
> > >
> > > Thanks for your careful and valuable comments. We will explain your concerns point by point.
> > >
> > > Q1: The inconsistent results of teacher and student detectors
> > >
> > > We conduct experiments according to the official mmdet 1.2 version. We carefully checked the code of the experiment and ensured the accuracy of the experiment.
> > > Our method may get better localization accuracy. Although the effect of A50, AP75 is not obvious, it can still improve mAP (50-95). In addition, Our method has a significant improvement on large objects. Chapter 4.4.4 in the paper also explained that based on the pr curve, it has a better lifting effect on large objects.
> > >
> > > Q2: the longer training
> > >
> > > We conduct experiments on the Retina-Res50 without distillation on mmdetection 2. It can be seen that as the training time increases, the baseline student detector accuracy decreases, which may indicate that it has overfitted.
> > >
> > > | without_distillation | mAP  | AP50 | AP75 | APS  | APM  | APL  |
> > > |----------------------|------|------|------|------|------|------|
> > > | 1x                   | 36.5 | 55.4 | 39.1 | 20.4 | 40.3 | 48.1 |
> > > | 2x                   | 37.4 | 56.7 | 39.6 | 20   | 40.7 | 49.7 |
> > > | 3x                   | 36.5 | 55.6 | 38.6 | 19.4 | 39.3 | 49.6 |
> > >
> > > However, the effect of the student network distilled by our method has been further improved.
> > >
> > > | with_FRS  | mAP  | AP50 | AP75 | APS  | APM  | APL  |
> > > |-----------|------|------|------|------|------|------|
> > > | 1x        | 39   | 58.1 | 41.8 | 21.8 | 42.8 | 52   |
> > > | 2x        | 39.7 | 58.6 | 42.4 | 21.8 | 43.5 | 52.4 |
> > > | 3x        | 39.9 | 58.7 | 42.8 | 22.1 | 43.9 | 53.1 |
> > >
> > > Q4: The unclear sentence
> > >
> > > Thank you for your suggestion, we will revise it in the paper

---

> > > > ### Comment · Reviewer_iJCP · 2021-09-02
> > > > **Comments**
> > > >
> > > > Thanks authors for further clarification! Now I am confident on comparison with the baselines and other existing approaches.

---

### Official Review · Reviewer_FmCT · 2021-07-17

**Rating:** 7
**Confidence:** 5

**Summary:**

This paper proposes to perform knowledge distillation on object detectors with feature richness. They use the feature-richness score to decide how to choose the important features. Experiments and qualitative analysis on COCO have been provided to evaluate their method.

**Limitations And Societal Impact:**

Please refer to main review.

**Main Review:**

Originality:  There have already been some methods that try to distill knowledge outside the anchors such as [1] and [2] so this is not a very novel topic. The authors propose to use the feature richness score to decide which region to be distilled, which is a new solution for this topic.

Quality:  The author does not provide enough comparison with some very related works such as [1]. Comparisons on both experiments and methodology should be provided. 2. the experiments are also not enough. More detectors such as DERT and more datasets such as Cityscapes should be considered.  3. Analysis in 4.4.3 is interesting.

Clarity. The writing of this paper is ok. But some figures (fig5, fig6, fig7) are blurry. Authors should provide figures in vectorgraph. And the fonts in these figures are too small.

Significance. The results of this paper are not very significant compared with previous papers such as [1].

[1] Improve Object Detection with Feature-based Knowledge Distillation: Towards Accurate and Efficient Detectors, ICLR2021
[2] General instance distillation for object detection, CVPR2021

------------
After Rebuttal, Rate 6 -> 7.


**Time Spent Reviewing:**

3

---

> ### Author Response · Authors · 2021-08-10
> **Response**
>
>
> We would appreciate your advice for the further improvement of our work. We would like to address your concerns one by one. Any further discussion would be appreciated.
>
> **Question1. Comparisons on both experiments and methodology**
>
> **Comparison on Experiments**
>
> we conduct experiments on RetianNet on MS COCO2017, and compare with [1] and [2].
>
> **Setup**: For fair comparison, we run the experiment using the teacher and student model of the same accuracy with [1] and [2], respectively. Therefore, the implementation of [1] is based on the mmdetection 2.0 framework, while the implementation of [2] is based on the mmdetection 1.2, as mentioned in "implementation detail" sections in [1] and [2].
>
> **Results** The results are as follows: Our method surpasses [1] by 0.5% mAP, and surpasses [2] by 0.2% mAP. It can be seen that our method has achieved a better improvement effect.
>
> |                         | mAP  | AP50 | AP75 | APS  | APM  | APL  |
> |-------------------------|------|------|------|------|------|------|
> | Retina-Res50(student)   | 37.4 | 56.7 | 39.6 | 20   | 40.7 | 49.7 |
> | Retina-ResX101(teacher) | 40.8 | 60.5 | 43.7 | 22.9 | 44.5 | 54.6 |
> | [1]                     | 39.6 | 58.8 | 42.1 | 22.7 | 43.3 | 52.5 |
> | ours                    | 40.1 | 59.5 | 42.5 | 21.9 | 43.7 | 54.3 |
>
> |                        | mAP  | AP50 | AP75 | APS  | APM  | APL  |
> |------------------------|------|------|------|------|------|------|
> | Retina-Res50(student)  | 36.2 | 55.8 | 38.8 | 20.7 | 39.5 | 48.7 |
> | Retina-Res101(teacher) | 38.1 | 58.3 | 40.9 | 21.2 | 42.3 | 51.1 |
> | [2]                    | 39.1 | 59   | 42.3 | 22.8 | 43.1 | 52.3 |
> | ours                   | 39.3 | 58.8 | 42   | 21.5 | 43.3 | 52.6 |
>
> **Comparison on Methodology**
>
> Our method show obvious differences with [1] and [2] in methodology.
> The feature masks used to guide student detector in [1] are gained from the feature maps of both teacher detector and student detector with the help of attention. In contrast, the feature masks of our method are obtained from the classification score of teacher detector to approximate the probability that the features are objects, i.e. the feature richness. Thus, in motivation, our method has a clear insight and interpretability of computing feature richness compared with [1]. With the effectiveness of feature richness, our method can effectively retrieves the important features outside the bounding boxes and removes the detrimental features within the bounding boxes. Moreover, our method using classification score of teacher detector can obtain information more relevant with semantic which brings more advantages of recognizing the objects.
> Moreover, the feature masks learned in [2] are boundingbox-wise and binary, while our method generates feature masks are pixel-wise and fine-grained. Therefore, our method show higher ability to elaborately choose important features that improve generalized detectability during distilling.
> Furthermore, our method is quite clean and simple, but effective. Our method need less hyperparameters and is easier to implement.
>
> **Question2. Experiments on more models and datasets**
>
> **More dataset**: Pascal VOC dataset is widely used in object detection, so we conduct the extend experiment results on Faster R-CNN and RetianNet on Pascal VOC.
> The ResNet50 based FasterRCNN surpasses the baseline by 1.7% mAP. The ResNet50 based on RetinaNet gain 2.9% mAP. It can be seen that our method has also achieved significant improvements on other data sets.
>
> | Model                  | AP50 |
> |------------------------|------|
> | faster-Res101(teacher) | 81.3 |
> | faster-Res50(student)  | 79.5 |
> | ours:                  | 81.2 |
> | gains:                 | 1.7  |
> | Retina-Res101(teacher) | 80.1 |
> | Retina-Res50(student)  | 77.3 |
> | ours                   | 80.2 |
> | gains                  | 2.9  |
>
> **More detector**: Considering the DETR detector requires too long training time and we need to train both the teacher detector and the distilled student network, and the time is far from enough during rebuttal period (at least 8 days). Therefore, we use YOLOF detector instead. YOLOF[1] is the latest detector without FPN module, so it is different in construction from the detector in our paper. Therefore YOLOF is suitable to evaluate the generalization of our methods. The following table shows the experiment results on YOLOF on MS COCO2017. YOLOF with ResNet50 achieves 39.3% in mAP, meanwhile surpasses the baseline by 1.8% mAP.
>
> | Model                 | AP   | AP50 | AP75 | APS  | APM  | APL  |
> |-----------------------|------|------|------|------|------|------|
> | YOLOF-Res50(student)  | 37.5 | 57   | 40.4 | 19   | 42   | 53.2 |
> | YOLOF-Res101(teacher) | 39.7 | 59   | 42.9 | 20.5 | 44.3 | 55   |
> | ours                  | 39.3 | 58.5 | 42.5 | 20.5 | 44   | 54.5 |
>
> **Question3 About Figure.**
>
> Thanks for your suggestion, we will carefully revise the manuscript to make it easier to read.
>
> [1] Improve Object Detection with Feature-based Knowledge Distillation: Towards Accurate and Efficient Detectors, ICLR2021
>
> [2] General instance distillation for object detection, CVPR2021

---

> > ### Comment · Reviewer_FmCT · 2021-08-28
> > **After Reading Author Response**
> >
> > Thanks for the response from authors.
> > I think their response has solved most of my concerns. Thus, I tend to increase my rate from 6 to 7.

---

### Decision · Program_Chairs · 2021-09-27

**Decision:**

Accept (Poster)

**Comment:**

The paper presents a method (Feature-Richness Score) to choose important features to distill for object detection, leveraging a feature pyramid network. There is significant prior work in the domain but the method is different enough. The experimental evaluation is convincing as it encompasses several detectors, backbones, and datasets. A limit of the method is that it does not work out of the box with approaches without FPN (e.g. DETR). Overall, the paper can be improved easily and the authors' response convinced 3 reviewers (out of 4) to raise their score by one point. The outcome makes this paper fitting as poster at NeurIPS.